# Discussing Personalized Prognosis Empowers Patients with Amyotrophic Lateral Sclerosis to Regain Control over Their Future: A Qualitative Study

**DOI:** 10.3390/brainsci11121597

**Published:** 2021-11-30

**Authors:** Remko M. van Eenennaam, Loulou S. Koppenol, Willeke J. Kruithof, Esther T. Kruitwagen-van Reenen, Sotice Pieters, Michael A. van Es, Leonard H. van den Berg, Johanna M. A. Visser-Meily, Anita Beelen

**Affiliations:** 1Department of Rehabilitation, Physical Therapy Science and Sports, UMC Utrecht Brain Center, University Medical Center Utrecht, 3508 GA Utrecht, The Netherlands; r.m.vaneenennaam@umcutrecht.nl (R.M.v.E.); L.L.S.Koppenol@umcutrecht.nl (L.S.K.); W.Kruithof@umcutrecht.nl (W.J.K.); E.T.Kruitwagen@umcutrecht.nl (E.T.K.-v.R.); J.M.A.Visser-Meilij@umcutrecht.nl (J.M.A.V.-M.); 2Center of Excellence for Rehabilitation Medicine, UMC Utrecht Brain Center, University Medical Center Utrecht, and De Hoogstraat Rehabilitation, 3508 GA Utrecht, The Netherlands; 3Basalt Rehabilitation, 2543 SW The Hague, The Netherlands; s.pieters@basaltrevalidatie.nl; 4Department of Neurology, UMC Utrecht Brain Center, University Medical Center Utrecht, 3508 GA Utrecht, The Netherlands; M.A.vanEs@umcutrecht.nl (M.A.v.E.); L.H.vandenBerg@umcutrecht.nl (L.H.v.d.B.)

**Keywords:** amyotrophic lateral sclerosis, motor neuron disease, prognosis, palliative care, quality of life, qualitative study

## Abstract

The ENCALS survival prediction model offers patients with amyotrophic lateral sclerosis (ALS) the opportunity to receive a personalized prognosis of survival at the time of diagnosis. We explored experiences of patients with ALS, caregivers, and physicians with discussing personalized prognosis through interviews with patients and their caregivers, and in a focus group of physicians. Thematic analysis revealed four themes with seven subthemes; these were recognized by the focus group. First, tailored communication: physician’s communication style and information provision mediated emotional impact and increased satisfaction with communication. Second, personal factors: coping style, illness experiences, and information needs affected patient and caregiver coping with the prognosis. Third, emotional impact ranged from happy and reassuring to regret. Fourth, regaining control over the future: participants found it helpful in looking towards the future, and emphasized the importance of quality over quantity of life. Personalized prognosis can be discussed with minimal adverse emotional impact. How it is communicated—i.e., tailored to individual needs—is as important as what is communicated—i.e., a good or poor prognosis. Discussing personalized prognosis may help patients with ALS and their caregivers regain control over the future and facilitate planning of the future (care). For many patients, quality of life matters more than quantity of time remaining.

## 1. Introduction

Life expectancy in amyotrophic lateral sclerosis (ALS) varies greatly, ranging from months to over 10 years from disease onset [1]. When receiving their diagnosis, most people with ALS are informed that, although variable, average survival is 3 to 5 years from disease onset [2]. The Personalized ENCALS survival prediction model for ALS allows physicians to estimate a more personalized prognosis—i.e., expected survival in individual patients from symptom onset to death, tracheostomy, or non-invasive ventilation for more than 23 h per day—at diagnosis [3]. This is especially relevant, since the prediction model shows that in about 60% of patients, average survival is either an overestimation or underestimation, which can have a negative impact on the emotional wellbeing of patients and their families [4].

Adequate and accurate discussion of prognosis in life-limiting neurological disorders is acknowledged as being important in supporting patient-centered care [5,6]. Many clinical decisions and patients’ choices cannot be fully informed unless the prognosis is considered [7]. However, there are important barriers to prognostic disclosure. Physicians may not feel confident in delivering bad news and may fear a negative impact on patient’s hope or mental wellbeing; this may even cause physicians to avoid discussing prognostic information altogether [6,8,9]. Furthermore, filling out a prediction model, interpreting outcomes, and communicating numerical estimates like a prognosis in a way that is easily understandable for patients, can also seem challenging [10,11,12]. To date, there have been no studies focusing on prognostic disclosure in neurological disease. However, studies in other life-limiting diseases, predominantly terminal cancer, show that prognosis can safely be discussed with patients and their caregivers as long as communication is tailored to their preferences and needs; this may even benefit patient decision-making and planning for the future, and provide a sense of control [13,14].

We developed a communication guide to help physicians overcome barriers to prognostic disclosure and support them in discussing the personalized prognosis in ALS with care, and tailored to patient and caregiver needs [13]. However, given the absence of research on the (emotional) impact of prognostic disclosure in life-limiting neurological disorders and ALS, physicians may find it difficult to discuss life expectancy [9]. Qualitative research is specifically suited to obtain deeper insight into the experiences of participants involved [15]. In the present study, we explored experiences of people with ALS, their caregivers, and their physicians when discussing the personalized prognosis based on the ENCALS prediction model [3].

## 2. Methods

### 2.1. Design

This is a qualitative study using thematic analysis; data are reported in accordance with the Consolidated criteria for reporting qualitative research (COREQ) checklist (Appendix A) [16,17].

### 2.2. Setting

After receiving the diagnosis of ALS from the neurologist, most patients are referred to one of 38 multidisciplinary teams responsible for ALS care in The Netherlands where ALS care is part of (rehabilitation) palliative care. ALS care teams are coordinated by a rehabilitation physician. Three ALS care teams were involved in the recruitment for this study, one associated with a university hospital (UMC Utrecht) and two with rehabilitation centers (Basalt Den Haag and Basalt Leiden).

### 2.3. Participants

#### 2.3.1. Patients with ALS and Their Caregivers

Patients and their caregivers were eligible for inclusion if the personalized prognosis based on the ENCALS prediction model had been discussed with them by their neurologist or rehabilitation physician within six months of the diagnosis of ALS. The ENCALS prediction model, based on data from over 11,000 patients with ALS in population-based registers, allows physicians to estimate the personalized prediction of survival at diagnosis. The model is based on eight factors: age, El Escorial classification, site of onset, vital capacity, genetic status for C9orf72 expansion, diagnostic delay, cognitive status and functional score [3]. Physicians were encouraged to use the communication guide to support them in discussing the personalized prognosis [13]. Patients were recruited by physicians at three ALS care teams in The Netherlands (UMC, Utrecht; Basalt Den Haag; Basalt Leiden) using convenience sampling. Interested patients and their caregivers were sent an information leaflet on the study and contacted by one of the researchers (RvE, LK) to inform them about the study. After written consent had been provided, a date and time convenient to the participants was agreed upon for the interview within one month after discussing the personalized prognosis. Patients with PMA or PLS were not eligible for inclusion because the ENCALS prediction model is only calibrated for patients with ALS [3]. Patients with ALS and frontotemporal dementia (FTD) were also included in cases where the personalized prognosis was discussed with the caregiver.

#### 2.3.2. Physicians

All physicians who discussed the personalized prognosis and were involved in the recruitment of patients and caregivers for this study were invited to participate in a focus group.

### 2.4. Data Collection

#### 2.4.1. Patients and Caregivers

Semi-structured interviews with detailed probes were conducted by two researchers (RvE, LK) not involved in the care of the patients. Interviews were directed by an interview guide (Appendix A). RvE has been trained to conduct qualitative research and this is his third qualitative study. LK has been coached and supervised in the conduction of interviews and qualitative analysis by RvE. Both RvE and LK were supported by a senior researcher with extensive experience in qualitative research (AB). The interview guide was formulated by two researchers (RvE, AB) and based on a literature review which was performed as part of an earlier study on the development of a communication guide [13]. Interview topics included information needs [18,19,20,21,22,23,24,25,26], difference in experiences between patients and caregivers [19,21,22,24,27], emotional impact and hope [18,24,26,28,29], and satisfaction with prognostic disclosure [18,20,23,24,25,28,30]. Taking patient preferences into account, the interview was face-to-face at the ALS clinic or the home of the patient (pre-COVID-19) and recorded via telephone or video-consultation.

At the start of the interview, participant characteristics were registered (gender, age, level of education, and relationship of caregiver to patient). During the interview, patients and caregivers were asked to elaborate on their experiences discussing the personalized prognosis: how and when this was discussed, the impact (emotional or otherwise), and their satisfaction with the consultation including their suggestions for improvement. Participants were offered a transcript of the interview to make corrections and additions if needed (member check).

#### 2.4.2. Physicians

A focus group of physicians was led by two trained researchers (RvE, AB) and was directed by an interview guide (Appendix A); LK, present as observer, made field notes. The focus group was recorded via video-consultation. Physicians were asked to elaborate on their experiences discussing personalized prognosis with patients with ALS and their caregivers, and to reflect on the emerging themes from the interviews (with patients and caregivers).

### 2.5. Data Analysis

#### 2.5.1. Patients and Caregivers

Interviews were transcribed verbatim, anonymized, and analyzed by two researchers (RvE, LK) using an inductive approach. The process of data collection and analysis was iterative, proceeding simultaneously to provide the opportunity for important emerging topics to be incorporated into subsequent interviews. Inclusion proceeded until data saturation was reached, i.e., when no new themes emerged during the last three interviews [31]. First, transcripts were read to become familiar with the narrative. Second, the texts were broken down into fragments based on their content and coded independently by two researchers (RvE, LK) in NVIVO 12 (NVivo Qualitative Data Analysis Software; v. 12.6) using open coding [32]. Resulting codes and discrepancies were compared and discussed to enhance credibility of the results and minimize interpretation bias. Third, after every 4-5 interviews, existing codes were evaluated by the research team (RvE, LK, AB, WK) and, where necessary, recoded. Fourth, codes were sorted and categorized into overarching themes and subthemes using thematic analysis [16]. A descriptive summary of each theme was written, and quotes were linked to the themes by one researcher (RvE) to express the essence of the content; themes were discussed by the research team (RvE, LK, AB, WK, EKR, MvE).

#### 2.5.2. Physicians

The focus group was transcribed verbatim and analyzed by two researchers (RvE, LK) similarly as described above. The goal of the focus group was to explore physician experiences discussing personalized prognosis and to discuss the most important patient and caregiver themes.

## 3. Results

### 3.1. Participants

A total of 16 interviews were performed in 14 cases, involving thirteen patients and ten caregivers (eight partners and two adult children), between October 2019 and May 2021 (Table 1). Data saturation was reached after we had interviewed nine patients and six caregivers in ten cases; the recording of one interview failed due to technical issues (C6) and could not be analyzed. Four rehabilitation physicians and one neurologist were included in the focus group (Table 2). Included patients represented different age and prognostic groups (except for the very short prognostic group); most participants had received a high level of education; five an intermediate level. In one case, only the caregiver was interviewed because the patient had FTD (C1); children were interviewed separately. Interviews took between 21 and 66 min; the focus group with physicians lasted 60 min.

### 3.2. Patient and Caregiver Themes

The analysis of the interview data revealed four overarching themes with seven subthemes (see Figure 1).

#### 3.2.1. Tailored Communication


**Communication style**


Patients and caregivers emphasized the importance of a person-centered communication style tailored to their emotional needs (quotes 1, 2 in Table 3). When the physician’s style did not meet their preferences, this led to dissatisfaction (quote 3).


**Information provision**


They also expressed their satisfaction when prognostic information was tailored to their needs (quotes 4, 5). Empathetic, tailored communication did not have to take up much time, however, patients and caregivers also emphasized the importance of adequate preparation by the physician (quote 6). Patients reported inconsistency between the information provided on the average life expectancy in ALS at diagnosis and the personalized prognosis they received later. This inconsistency could increase the negative emotional impact of bad news (quote 7, 25). Generally, the personalized prognosis was discussed as a *range* and the inherent statistical uncertainty was emphasized by the physician. The better end of this range could provide a measure of hope (quote 8). However, it could also cause confusion if the underlying prediction model and range were insufficiently explained (quote 5).

#### 3.2.2. Personal Factors

The subthemes coping style, illness experiences, and information needs were strongly interrelated with each other, but generally not with role (patient or caregiver), gender, or prognosis.


**Coping style**


Often patients described themselves as down-to-earth and displayed an active, problem-focused coping style. They said that this coping style and knowing their personalized prognosis helped them to confront and accept their prognosis and start planning for the future instead of dwelling on their emotions (quotes 9, 10 in Table 4). Older patients and caregivers (i.e., over 65 years old) often reflected with satisfaction on a long and fulfilled life making it easier for them to accept and cope with their prognosis, regardless of whether it was good or bad news (quotes 11, 12). One patient at first exhibited an avoidant coping style, due to the death of his sibling from ALS, but later he did want to know his personalized prognosis (quote 17). Due to inconsistency of information, another patient responded with regret and denial because his personalized prognosis turned out to be bad news (quote 25). The coping styles of caregivers also varied. Some had difficulty coping with the situation (quote 10) and, as a result, did not always wish to participate in the interviews, but others wanted information about the personalized prognosis in order to regain some measure of control (quotes 13).


**Illness experiences**


Patients reported that, prior to prognostic disclosure, they already had an expectation about what their life expectancy would be, based on their experience with the rate of disease progression. Some described the personalized prognosis as reassuring (quote 14), whereas others questioned its added value (quotes 15). A number of caregivers, however, said that this information was important to them because it confirmed what their partner already felt (quote 15). When the prognosis did not match the patient’s feelings, they described this as surreal (quote 16). Prior illness experiences with ALS within the family and other diseases could affect how participants coped with prognostic disclosure (quotes 17, 18).


**Information needs**


Information needs for the personalized prognosis varied between participants. Some said they wanted as much detailed information as possible about their personalized prognosis and the underlying model (quote 19), whereas others preferred a more general indication (quote 20). Although in most cases the physician broached the topic, sometimes patients and caregivers took the initiative and requested information about the prediction model (quote 19). Some caregivers found it harder to cope with the situation and were sometimes taken aback by the patients’ desire for information on the personalized prognosis (quote 19, 10), others needed clarity (quote 21, 15, 28). There were also reports by patients and caregivers that their need for information about functional prognosis was not met (quote 22).

#### 3.2.3. Emotional Impact

The emotional impact of prognostic disclosure ranged from happy and reassuring to regret. Patients and caregivers said that good news about their personalized prognosis made them happy and gave them a feeling of having time and peace (quote 23 in Table 5) but emphasized that the communication style mediated the emotional impact (quotes 1, 2). A shorter than average prognosis could be a more difficult message to cope with for participants (quote patient 24) and one patient expressed regret over agreeing to discuss his personalized prognosis (quote 25). Many of the older patients, however, were more accepting, and some described the limited life expectancy as a relief (26, 18). Regardless of good or bad news, none of the patients said it caused them anxiety or to lose hope and most patients were satisfied with the discussion of their personalized prognosis.

#### 3.2.4. Regaining Control over the Future


**Helpful in looking towards the future**


The majority of patients and caregivers stated that discussing their personalized prognosis was helpful as it provided clarity and alleviated uncertainty (quotes 27, 28, 21). They told us how this information helped them regain some measure of control, enabling them to redefine and plan for the future (quotes 29, 10), including future care (quotes 11, 15), as well as what would happen after the patient’s death (quote 30). Also, this knowledge helped participants make the most of the time they had left (quotes 28, 11).


**Quality over quantity**


Without being prompted to discuss the topic, the majority of patients, and some caregivers, emphasized the importance of quality of life over the quantity (quote 22). For them a good quality of life meant allowing them to engage in meaningful activities, to communicate with loved ones and friends, and to express their own will (quotes 31). A number of patients divided their remaining time into a “acceptable” part with a satisfactory quality of life and contrasted this with a “bad” part while reflecting on taking control over the end-of-life (quotes 31, 32, 33). Although a few patients and caregivers expressed the hope of being on the ‘good side’ of the prognosis, they hoped more often for a satisfactory quality of life for as long as possible (quotes 33, 34).

### 3.3. Physician Focus Group

Patient and caregiver themes described above were also recognized by the physician focus group (quotes p1–p9 in Table 6). Analysis of the focus group revealed potential benefits of discussing personalized prognosis and barriers.


**Benefits**


Physicians agreed that discussing personalized prognosis is not that different from other difficult conversations about bad news in ALS (quote p10). The two more experienced physicians said that even with many years of experience, the prediction model provided valuable information because they were sometimes surprised by the outcome (quote p11). Additionally, discussing personalized prognosis can enhance more personalized care (quote p12).


**Barriers**


Physicians underscored that preparation, interpreting the outcomes, especially in the case of missing (e.g., the outcome of genetic testing for C9orf repeat expansion) and uncertain data (e.g., date of symptom onset), requires time and effort, as does explaining the model, and responding to emotions (quote p13). They also discussed the dilemma of missing or unclear variables when filling out the model and how this can result in different outcomes (quote p14). Another topic that was discussed at length concerned the difficulty of prognostic disclosure with patients with a language barrier, which could be further confounded by a different cultural background (quote p15).

## 4. Discussion

This is the first study investigating experiences of patients, caregivers, and physicians when discussing personalized prognosis of survival based on a prediction model in neurological disease. Our study shows that personalized prognosis can be discussed with patients with ALS and their caregivers without negative impact, provided the physicians tailor communication to individual needs and preferences. Personalized prognosis may help patients and their caregivers regain control over the future, and can facilitate future planning, where maintaining quality of life is more important than survival time.

Most of the experiences of patients with ALS and their caregivers in our study are in agreement with studies on prognostic disclosure in other life-limiting diseases, and show that concerns about an adverse impact on psychological wellbeing of patients and caregivers are unwarranted [8,13,14]. Indeed, studies show that an unfulfilled desire for a more personalized prognosis can cause frustration and distress for patients and caregivers [24,33,34], whereas patients and caregivers in our study reported that discussing personalized prognosis can alleviate uncertainty. Participants in our study differed in their information needs, some desiring a very detailed explanation of their personalized prognosis and prediction model and others wanting a more general indication of their life expectancy. All participants agreed about the importance of empathetic and honest communication. Our study and others show that exploring and tailoring prognostic disclosure to the emotional and information needs of patients and their caregivers mediates the emotional impact, supports acceptance, and improves satisfaction with the communication [14,18,24,27,29].

Prognostic disclosure may also promote acceptance and coping by supporting patients with life-limiting diseases and their caregivers to redefine and plan for their future, including future care, thus allowing them to focus on the present and their quality of life [29,30,35]. Prognostic awareness also supports patients in planning future care together with their caregivers and physicians, and allows for clinical and end-of-life decision-making that is better aligned with patients’ values and preferences [7,18,24]. This has been associated with a better quality of life, especially nearing end-of-life [36,37]. In ALS specifically, this may help patients, caregivers, and ALS care teams to ‘stay one step ahead’ when planning future care and end-of-life [38]. All of this can help patients regain a sense of control [30], especially relevant to patients with ALS who, from diagnosis, are confronted by the prospect of ever-present and increasing loss [33,39,40,41].

Coping style and illness experiences are important personal factors that mediate the acceptance and outcome of prognostic disclosure. Similar to our results, patients with advanced, incurable cancer preferring a more active coping style were more likely to want information about their prognosis and to use this in planning their future and future care [30]. Additionally, older patients may find it easier to accept a poor prognosis, because they can reflect on a long, fulfilled life [30]. Before discussing their personalized prognosis, many patients in our study already anticipated this on the basis of the changes they did or did not feel in their body. Living with a disease and experiencing symptoms may promote coping and acceptance, and limit the emotional impact of a poor prognosis [30]. Some patients and caregivers in our study reported diverging prognostic information needs depending on their coping style and acceptance of the disease [30]. A number of studies in ALS [42,43] and other life-limiting diseases [21,24] report that some caregivers may have a stronger desire for prognostic information than the patient. Our study suggests that, in addition to the need to plan future care and for the time after the patient’s death, some caregivers may have a need for prognostic certainty, because they are not experiencing the rate of deterioration and cannot feel what the patient is feeling.

Physicians in our focus group agreed that, in many ways, discussing personalized prognosis is not so different from other bad news conversations. Communication guidelines [44,45] can support physicians in tailoring prognostic disclosure to individual patient needs. Discussion of prognosis itself takes little time but should be placed in the broader, holistic context of the patient’s values and dignity, their perspective on the future, goals of care and treatment options, and quality of life.

Our results differ from those in other life-limiting diseases, in particular cancer, with respect to the relationship between prognosis and hope. Hope is acknowledged as essential in life-limiting disease and ranges from the hope to be cured, to having a longer life, a good life, to a good death. In cancer there is often a realistic hope for a cure and even in the palliative phase, cancer patients often hold out the hope of remission or a cure [28,46]. When diagnosed with ALS, there is no prospect of a cure or remission, and patients are immediately referred to palliative care. As a result, shortly after diagnosis, most patients with ALS and their caregivers redefine their hope from hope of a cure to focusing on hoping for a meaningful life [47]. Patients in our study often focused more on the hope of maintaining a satisfactory quality of life rather than the quantity of time left.

While reflecting on their quality of life and remaining time, patients in our study also talked about taking control of the end-of-life through euthanasia. After receiving their diagnosis, most patients with ALS in the Netherlands inquire about the possibility of euthanasia [2]. In other countries, many ALS patients consider hastening their death and would welcome the opportunity to discuss this topic with their physician; however, this often does not happen [48,49]. Reasons to consider or actually hasten death include loss of autonomy and dignity, disability impairing quality of life, and a desire to control the end-of-life [49,50,51]. Better prognostic awareness may support advance care planning and end-of-life decision-making, which can relief anxiety, provide a sense of control, support hope, and facilitate both quality of life and quality of dying when the disease becomes too much to bear [52].

The use of prediction models to predict survival and support decision-making is on the rise [10,12]. Physicians in our focus group agreed that filling out the model, interpreting outcomes, and communicating estimated survival (and its uncertainty) in an easily understandable manner for patients takes some time, but after a small learning curve, it was no more difficult or stressful than other bad news conversations in ALS. The accuracy of a prediction model can be impaired by missing or unclear variables [53]. The two most important predictors for the ENCALS prediction model and most at risk of uncertainty are date of onset and vital capacity [3]. If these data are uncertain, physicians should consider the impact on the outcome and decide whether it is feasible to discuss the personalized prognosis. Physicians in our focus group were hesitant to discuss the personalized prognosis with patients with a language barrier and those from non-western cultures. However, non-western studies show a positive association between prognostic disclosure and quality of life; preferences about prognostic disclosure may differ among and within non-western cultures [54,55]. It is recommended that physicians approach all patients, regardless of their cultural background, in the same way, by exploring their preferences and needs regarding this topic [44]. The above-mentioned topics have already been incorporated in our communication guide [13].

This and other studies show that at least some patients with ALS and their caregivers would like to receive a more personalized prognosis than the average life expectancy of all patients [3,33,42,56]. Some of the participants in our study took the initiative to ask their physician about the prediction model and their personalized prognosis after having read or heard about it. However, information discrepancy between the average life expectancy discussed at diagnosis and personalized prognosis may cause dissatisfaction and be detrimental to the patient-physician relationship [18]. Proper discussion of personalized prognosis in the broader context of ALS care trajectory may not always be possible at diagnosis due to time restraints and the shock of diagnosis-limiting information retention. However, besides discussing the fact that ALS is a lethal disease with a limited, but variable life expectancy, neurologists in the Netherlands should inform patients that they can be offered a more personalized prognosis at a later date, either in a second consultation with the neurologist or with the rehabilitation physician.

Finally, to facilitate uptake of discussing personalized prognosis in ALS, the online ENCALS prediction model could be made more user-friendly by integrating instructions to handle missing and unclear variables, and recommendations on how to discuss the outcome; also, the model should be validated in more populations.

### Strengths and Limitations

An important strength of this study is the robust methodological design with two independent coders, and multiple viewpoints from patients, caregivers, and physicians. A possible limitation is that most participants had received a high level of education and none a lower level. We also lack the perspective of patients with a very short prognosis. However, due to their fast rate of disease progression, these patients are probably already aware of their poor prognosis and a personalized prognosis would offer very limited additional benefit.

## 5. Conclusions

Personalized prognosis can be discussed with patients with ALS who want information about their individual life expectancy, and with their caregivers, with minimal adverse emotional impact. For the emotional impact, how the message is communicated—i.e., a person-centered communication tailored to their emotional and information needs—is as important as what is communicated—i.e., a good or poor prognosis. Discussing personalized prognosis shortly after diagnosis may help patients with ALS and their caregivers regain control over the future and can facilitate planning (of the future) including future care. For many patients, quality of life matters more than quantity of time remaining.

## Figures and Tables

**Figure 1 brainsci-11-01597-f001:**
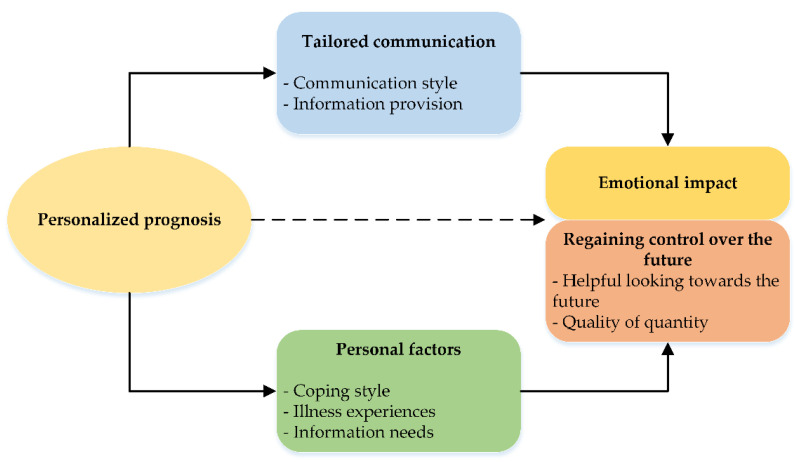
Overarching themes and subthemes on patient and caregiver experiences with discussing personalized prognosis in amyotrophic lateral sclerosis.

**Table 1 brainsci-11-01597-t001:** Description of cases.

Case	Participant	Sex	Age	Education	Initiative to Discuss	Prognostic Group	Location Interview	Participation
C1 *	Patient 1	Male	64	High	Physician	Long		No
	Partner 1	Female	50	Intermediate			Clinic (face to face)	Yes
C2	Patient 2	Female	57	Intermediate	Physician	Very long	Home (face to face)	Yes
C3	Patient 3	Female	69	High	Physician	Short	Home (telephone)	Yes
C4	Patient 4	Female	73	High	Patient-caregiver	Long	Home (face to face)	Yes (separate)
	Daughter 4	Female	49	High			Home (telephone)	Yes (separate)
C5	Patient 5	Male	71	Intermediate	Physician	Short	Home (telephone)	Yes
C6 **	Patient 6	Male	65	High	Physician	Long	Home (face to face)	Yes (together)
	Partner 6	Female	-	High			Home (face to face)	Yes (together)
C7	Patient 7	Male	59	High	Physician	Intermediate	Clinic (face to face)	Yes
C8	Patient 8	Male	52	High	Patient-caregiver	Long	Home (Video)	Yes
C9	Patient 9	Male	55	High	Physician	Long	Home (telephone)	Yes (together)
	Partner 9	Female	54	High			Home (telephone)	Yes (together)
C10	Patient 10	Female	68	High	Physician	Very long	Home (telephone)	Yes (together)
	Partner 10	Male	72	High			Home (telephone)	Yes (together)
C11	Patient 11	Male	56	Intermediate	Physician	Short	Home (Video)	Yes (together)
	Partner 11	Female	54	Intermediate			Home (Video)	Yes (together)
	Daughter 11	Female	24	High			Home (Video)	Yes (separate)
C12	Patient 12	Male	57	High	Physician	Short	Home (Video)	Yes (together)
	Partner 12	Female	47	High			Home (Video)	Yes (together)
C13	Patient 13	Male	79	High	Physician	Intermediate	Home (Video and telephone)	Yes (together)
	Partner 13	Female	81	High			Home (Video and telephone)	Yes (together)
C14	Patient 14	Female	77	High	Patient-caregiver	Short	Home (Video)	Yes (together)
	Partner 14	Male	81	High			Home (Video)	Yes (together)

* Patient has frontotemporal dementia; interview was only with the caregiver. ** Recording of the interview failed due to technical issues and could therefore not be included.

**Table 2 brainsci-11-01597-t002:** Characteristics of physicians.

Physician	Sex	Age	Years of Experience with ALS	Number of Times Physician Has Discussed Personalized Prognosis *	Medical Discipline
Physician 1	Female	53	15	15	Rehabilitation
Physician 2	Female	47	15	15	Rehabilitation
Physician 3	Female	34	5	10–15	Rehabilitation
Physician 4	Female	32	1.5	5	Rehabilitation
Physician 5	Male	30	3	10–15	Neurology

* Estimation by physician.

**Table 3 brainsci-11-01597-t003:** Patient and caregiver quotes on tailored communication.

Themes and Subthemes	Quotes
**Tailored communication**	
*Communication style*	Patient 4: “It was a pleasant conversation, yes a bit cheerful though. I was fine with it and we did leave there happy… First, that the life expectancy was obviously longer than we originally thought. And also just the way the [the rehabilitation physician] handled the situation, yes with humor… I [thought] it was special how [the rehabilitation physician did her] best to assess what type of person I am and how I’m handling it all. It was apparent that that was important to her.” **Quote 1. Patient 4 (73 years old); Long prognosis**
	Patient 8: “I think that is very important in a conversation like that that you are unburdened in the sense of … we are there to constantly assist you throughout this whole process and you are not alone… That combination of life expectancy combined with the fact that you are not facing it alone, I do find that essential. That combination, that gave me a sense of calm.” **Quote 2. Patient 8 (52 years old); Long prognosis**
	Patient 12: “It really just comes down to the person giving you the news… That might also be because this doctor is less empathetic than another doctor… I had the impression that she found it harder to tell me than I found dealing with it…” Partner 12: “We have no idea what’s on the computer because she’s looking at her computer screen and we’re sitting there… Either let us see what’s on your computer screen or turn off your screen and write it on a piece of paper.” **Quote 3. Patient 12 (57 years old) and Partner 12 (47 years old); Short prognosis**
*Information provision*	Patient 8: “We had a quick look at the screen together and I was able to get a look at the parameters… So that immediately gave me a sense of how that information is structured and what are, say, the key features… With that, to my mind, the matter was over and done with [laughs]… I think she [rehabilitation doctor] understood very well that I was interested, including in the scientific background of that life expectancy curve.” **Quote 4. Patient 8 (52 years old); Long prognosis**
	Partner 11: “Actually, the model is not clear to us… Yeah, and then when it’s said from 18 to 30 months that’s also, yeah, I think it’s almost like trying to read tea leaves. Patient 11: “I mean look, I’m pretty happy about it [consultation], but my wife and my daughter not so much [laughs].” Partner 11: “If she [the rehabilitation physician] herself already indicates that you have to look at that broadly then I think, so, is this false hope? False information? … I’m like yeah, but what are we taking with a grain of salt here? The 18 months or the 30 months, or the whole story?” **Quote 5. Patient 11 (56 years old) and Partner 11 (54 years old); Short prognosis**
	Daughter 11: “She said she was going to discuss it and then the computer didn’t work and then she had logged in somewhere else. And I think it’s pretty tough when you start giving information like that to someone, with a model like that and then that it’s not ready and then you’re waiting for it to be ready.” **Quote 6. Daughter 11 (24 years old); Short prognosis**
	Patient 7: “I went to [local hospital] first. That’s where I got the diagnosis: ALS. And, uh, yeah they were already talking about, well, several years… So then I was referred here [ALS Centre diagnosis day]. And then I was told 3 to 5 years. And yesterday [at the rehabilitation physician] … then it was 3 years. So, uh, that really has an impact… It’s gotten worse three times.” **Quote 7. Patient 7 (59 years old); Intermediate** **prognosis**
	Patient 8: “Here I was actually told three to five years, which already sounds a little better. And um, actually you don’t know anything then, because of course it’s a statistic, and then of course you have a spread and who knows, maybe I’m in the 96th percentile. And then I might end up with ten years. You never know.” **Quote 8. Patient 8 (52 years old); Long prognosis**

**Table 4 brainsci-11-01597-t004:** Patient and caregiver quotes on personal factors.

Themes and Subthemes	Quotes
**Personal factors**	
*Coping style*	Patient 7: “It was a bit intense at first, and also emotional. But then again, I’m so down-to-earth that yes, I resigned myself to it pretty quickly… [My wife], she’s a little more emotional than I am. She’s a little less down-to-earth… I accept that things are the way they are more easily.” **Quote 9. Patient 7 (59 years old); Intermediate prognosis**
	Patient 11: “I like things to be clear… then you can take action, do things you still want or take care of things… [With such a short life expectancy] you’re going to get started with things sooner and find things out a little faster to see what needs to get done… You distract yourself a little bit that way.” Partner 11: “Yeah, I was in denial. In particular, not being ready to get this news… I’m just like, we’ll see what’s coming and then we’ll just deal with it and I don’t need to know when that will be.” **Quote 10. Patient 11 (56 years old) and Partner 11 (54 years old); Short prognosis**
	Patient 13: “I’m not afraid to die, that’s a very important principle. I don’t think it’s time to die yet, but once you get to 80, we do say ‘up to 80 is wonderful, but 80 to 100 sucks’ [laughs]. Once you reach your 80th birthday, you’re increasingly faced with deterioration… It’s not dramatic that my life is finite.” Partner 13: “It’s just the way it is and I’m not going to worry about it… I want to take care of him for as much as I can and I will do that with love… So, we’ll just live for now and enjoy life every day.” **Quote 11. Patient 13 (79 years old) and Partner 13 (81 years old); Intermediate prognosis**
	Patient 14: “Our oldest son passed away very suddenly just before he turned 20. That’s the biggest disaster that can happen to you. After that all disasters pale in comparison… I’m 77 and I’ve had a very long life. A lot of people don’t even get that old. I have wonderful memories so it’s been nice.” **Quote 12. Patient 14 (77 years old); Short prognosis**
	Partner 12: “I am an extreme control freak [laughs] both in my work and in my personal life … and I don’t function as well when I know that there are unanswered [issues].” **Quote 13. Partner 12 (47 years old); Short prognosis**
*Illness experiences*	Patient 8: “I was like, ‘I’m going to make it those three years,’ because I’m actually feeling incredibly good right now. Very fit and spry. I actually have no complaints right now. So I feel like I can last a while. Well, that was kind of confirmed by this news. So in that sense, of course, that’s only positive.” **Quote 14. Patient 8 (52 years old); Long prognosis**
	Patient 14: “Well, I made an estimate myself by looking at how fast things are deteriorating and since that was pretty fast, I thought, well, it’s not going to take very long then. I’m not going to make it four years, but I probably won’t make it one year either… It makes little difference to me…” Partner 14: “I have now received confirmation of what [my partner] thought. So there’s also no reason to let anything give us false hope or anything… I have to arrange all kinds of things for the future. I actually need to stay just one step ahead of her disease, which is worsening, for example with aids. That’s why I wanted to know. It makes a big difference whether you have to take care of someone who has four years left or someone who has one year left.” **Quote 15. Patient 14 (77 years old) and Partner 14 (81 years old); Short prognosis**
	Patient 11: “You’re also sick, but you don’t have the idea yet of being so bad that within a short period of time you end up in such bad shape that all the muscles stop working… It’s like it’s so far from your daily life because you’re still so healthy.” **Quote 16. Patient 11 (56 years old); Short prognosis**
	Patient 9: “I had a brother who died [of ALS] within three-and-a-half months… Fear absolutely, it’s of course burying your head in the sand.” Partner 9: “The fear that it would be four months for you too… So on the one hand you were very afraid… And when it was done, … you got a completely different result and that was a relief.” **Quote 17. Patient 9 (55 years old) and Partner 9 (54 years old); Long prognosis**
	Patient 4: “I myself also cared for my mother with dementia and for a husband with Alzheimer’s for 10 years… In the case of Alzheimer’s, you have no idea how long it will last, and to what degree, and how it will all end. This is actually a pretty well-defined situation, clear cut, I would say. At a certain point it ends. Done… It doesn’t have to take very long for me though.” **Quote 18. Patient 4 (73 years old); Long prognosis**
*Information needs*	Patient 8: “So I asked that rehabilitation physician ‘let’s hear it’ [laughs]. Yeah! I was like, what do I have to lose? And that also came as a big shock to those around me, I think… I was mainly very curious about which of those five curves you can wind up in. And I was also a little curious to know why. I was raised in a very scientific way.” **Quote 19. Patient 8 (52 years old); Long prognosis**
	Patient 2: “Seven years. Two more years after all… He also said that it’s a slow variant. And so, with that my questions were basically answered [laughs].” **Quote 20. Patient 2 (57 years old); Very long prognosis**
	Patient 1: “I would also like to have clarity to have a certain grasp on things. Every time I think, okay this is it, then we’re already a step further… He’s obsessive-compulsive [due to frontotemporal dementia] and I have to deal with it 24/7. So I would like clarity.” **Quote 21. Partner 1 (64 years old); Long prognosis**
	Patient 13: “I don’t really want to know how long I have left to live, but rather … how long I will be able to function as I function now… And that’s a burning question: what will my life be like in a year’s time? Statistically speaking, I’m still alive, but what will my quality of life be then?” **Quote 22. Patient 13 (79 years old); Intermediate prognosis**

**Table 5 brainsci-11-01597-t005:** Patient and caregiver quotes on emotional impact & regaining control over the future.

Themes and Subthemes	Quotes
**Emotional impact**	
	Patient 4: “Three years seems like a very short time, but now you have a bit more space and that gives some room to breathe. It just gave us room to breathe.” Daughter 4: “It does feel like we were given some time in a way.” **Quote 23. Patient 4 (73 years old) and Daughter 4 (49 years old); Long prognosis**
	Patient 7: “If they have given me a diagnosis of 10 years, okay, that would have been nice. So this prognosis of three years, that makes it extra difficult. Definitely. Also in the whole processing of it… Yes, it’s a bit of [short pause] an emotional rollercoaster right now.” **Quote 24. Patient 7 (57 years old); Intermediate prognosis**
	Patient 5: “[I was] a bit confused. So I went back to [the diagnosis day at the ALS Centre] where they had given three to four years… How can she [rehabilitation physician] say two years? That’s a difference of almost half! … If I had known beforehand that the result would be so bad, I wouldn’t even have started. Because I would rather live with the thought of three to four years than two.” **Quote 25. Patient 5 (71 years old); Short prognosis**
	Patient 3: “Suppose I have another six months or so. Then we’ll have been so good to each other, it must hurt a lot less to say goodbye. And it’s easier for them [children], as well, not to see their mother deteriorate.” **Quote 26. Patient 3 (69 years old); Short prognosis**
**Regaining control over the future**	
*Helpful in looking towards the future*	Patient 2: “It’s just nice to know that I have some more time. You know, that does take away some of the uncertainty.” **Quote 27. Patient 2 (57 years old); Very long prognosis**
	Partner 12: “I find that I get a lot of peace from that, that I know … where I stand, where we stand as a family, and that we also have to make every day a celebration. Every day that [patient] is well, we have a party. Strange as it might be, we have no time left… So you just live a much more active lifestyle and you grab everything you can get your hands on … ” Patient 12: “Of course it [life expectancy] is a disappointment, but on the other hand it offers clarity. So you’re going to have to get up more focused every day with that knowledge.” **Quote 28. Patient 12 (57 years old) and partner 12 (47 years old); Short prognosis**
	Patient 7: “I have an appointment with the company doctor on Monday and I think I’m just going to say, ‘y’know, with this life expectancy, I just want to stop working’. I just want to spend time regularly with the grandchildren and with my wife…If you know this [life expectancy], then of course you aren’t completely in control, but you can start planning something. What I actually couldn’t do before, when I had just been diagnosed.” **Quote 29. Patient 7 (59 years old); Intermediate prognosis**
	Partner 12: “Both of us talk with social services in which [patient] talks up until death and I talk after the death … that I shouldn’t really be looking about afterwards, but rather that I should live more NOW, do things with [patient] now. And [patient] also gets advice to look further ahead, because that’s where [the question] arises for me, because how am I going to support my children or our children when he’s gone?” **Quote 30. Partner 12 (47 years old); Short prognosis**
*Quality over quantity*	Patient 8: “I especially hope that I will remain ambulant, that I can keep walking for example. And minimally to be able to use my hands even if they become weaker… That means that I can mail, so I can communicate, voice my own wishes. For me that is fundamental to quality of life, that you are able to communicate your own wishes… If that is not possible anymore, I think, life will end for me.” **Quote 31. Patient 8 (52 years old); Long prognosis**
	Patient 3: “Suppose it were a year and a half. Then I think I would divide it into a year and six months and I think that the last six months is no longer acceptable to me… So my life expectancy is then one and a half years minus half a year, let’s say. I’m just going to take charge of that myself… That does give me peace of mind.” **Quote 32. Patient 3 (69 years old); Short prognosis**
	Patient 7: “It’s not just life expectancy, it’s also when you look at ALS: how it progresses. Then the quality of life, that deteriorates rapidly… And I’m really going to look into euthanasia. Because I really don’t want to keep going until the very last moment… Look. I’ve resigned myself to the fact that it may be 3 years. Yeah, you hope that the quality of life will be good for a little while longer, or that it will be good for at least three years.” **Quote 33. Patient 7 (57 years old); Intermediate prognosis**
	Partner 11: “The result is still between 18 to 30 months, then you hope for 30, that’s the hope, yeah.”Patient 11: “Yeah, you just hope that you can stay mobile and do things normally for as long as possible.” **Quote 34. Patient 11 (56 years old) and Partner 11 (54 years old); Short prognosis)**

**Table 6 brainsci-11-01597-t006:** Quotes from physician focus group.

Themes and Subthemes	Quotes
**Tailored communication**	
*Communication style*	“And I think it’s very important HOW you discuss it with the patient, and that you feel how someone is receiving that message. Can someone accept that message, or are you just stirring up a lot of resistance? And if that’s the case, how can you change your tone of voice or the way you present something so that it is well understood; so that the patient and the partner or close relative who is there can go along with it?” **Quote p1. Physician 2**
*Information provision*	“Of course, you always try in a conversation to get a clear idea in advance of the degree to which both parties wish to have the conversation. And what their expectations are and what thoughts they have about it. Yes and you know, you do try to reflect back those emotions that you notice or feel or see.” **Quote p2. Physician 3**
	“But I also think it depends very much on how you tell people. If you just present that statistic not as fact and reality and truth, but just as very much the relativity of the statistic and that it does not come down to the month or the day.” **Quote p3. Physician 1**
*Coping style*	“Yes, my patients were also fairly accepting of the news… But it is also perhaps a selection of the population that wants to know, let’s say, because they already want to know. They’re curious and they may already have an expectation of where they fall under.” **Quote p4. Physician 5**
**Personal factors**	
*Illness experiences*	“So I’ve only discussed it twice and with both of them their reaction was actually ‘well, that’s the prognosis we were expecting’. So both of them weren’t that shocked by the news.” **Quote p5. Physician 4**
*Information needs*	“I also had a patient once who couldn’t live with the fact that he didn’t know [the personalized prognosis] … The fact that he knew the model was there, for him, made him really want to know as well.” **Quote p6. Physician 5**
**Emotional impact**	
	“With others, you notice a very emotional reaction they are really shocked by what the results of the prediction model are, and then there it is in writing in black and white or visible on the computer. And the picture you share with them then is often different from what they heard at [the diagnosis].” **Quote p7. Physician 2**
**Regaining control over the future**	
*Helpful in looking towards the future*	“And some patients say that they like to know where they stand so they can make a plan and think, I have more time or less time to plan my life further, and giving them something to hold on to.” **Quote p8. Physician 2**
*Quality over quantity*	“I’ve also had people who actually found it very interesting, but then turned around and said ‘but you still can’t tell me how it’s going to go’. So they actually found that much more interesting… Yes, that is much more relevant of course, so they already put the outcome and the conversation into perspective themselves.” **Quote p9. Physician 1**
**Potential benefits and barriers**	
*Benefits*	“I think that also the way you present it and also if you include the patient in it, then it doesn’t have to be more difficult than any other important subject, for example, discussing the limits of treatment… It’s mainly about being able to talk to people about what their future looks like, even if it’s shortened, and what they find important in the short time they have left. **Quote p10. Physician 2**
	“There have also been a few times when I thought, ‘oh, this is worse than I thought’, based on my clinical view.” **Quote p11. Physician 2**
	“If you take the time, you are talking about things that actually affect the patient deeply… And, that, I think is a very nice step towards very personal guidance… It can deepen your contact nicely, which is a nice basis for further conversations.” **Quote p12. Physician 1**
*Barriers*	“The preparation takes more time … [and] getting the concept right and explaining it well takes more time than getting the message across. And then it takes a lot of time to absorb the patient’s reaction and interpret it correctly.” **Quote p13. Physician 2**
	“Sometimes you really have doubts. I find, for example, in some patients, someone who has had cramps for five years or has had cramps all his life, has had cramps for four years and has had functional loss since December, explain to me when the symptoms started.” **Quote p14. Physician 1**
	“ALS patients … who either did not speak Dutch at all or where both patient and family only spoke English, I notice that I find this a complicated subject… That takes a lot of time … I don’t get started with that…. How I should and can discuss this clearly with non-native speakers, which also often involves a whole cultural problem.” **Quote p15. Physician 2**

## Data Availability

The data in this study, including the coding scheme, are available in Dutch on request from the corresponding author. The data are not publicly available due to privacy and ethical reasons.

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
