# Peer review of "Discussing Personalized Prognosis Empowers Patients with Amyotrophic Lateral Sclerosis to Regain Control over Their Future: A Qualitative Study"

_brainsci, 2021, doi:10.3390/brainsci11121597_

Round 1

Reviewer 1 Report

An interesting paper looking at patient experiences. Further information on the ENCALS predictive model is necessary within this paper. The greatest weakness is the number of assumptions made. The discussion and conclusions require further work.The case descriptions are interesting however incomplete with interpretation

Author Response

We thank the reviewer for their comments, which we have addressed in the revised manuscript as discussed below.

Comments to the Author

An interesting paper looking at patient experiences. Further information on the ENCALS predictive model is necessary within this paper.

  1. Further information on the ENCALS predictive model is necessary within this paper.

Response: We have added an explanation to the methods of what the ENCALS prediction model does: “The ENCALS prediction model, based on data from over 11,000 patients with ALS in population-based registers, allows physicians to estimate the personalized prediction of survival at diagnosis. The model is based on eight factors: age, El Escorial classification, site of onset, vital capacity, genetic status for C9orf72 expansion, diagnostic delay, cognitive status and functional score.” (see lines 84-89)

Discussion

  1. The greatest weakness is the number of assumptions made. The discussion and conclusions require further work. The case descriptions are interesting however incomplete with interpretation.

Response: We have interpreted the concerns of the reviewer as comments on the qualitative methodology. We have taken a number of steps to be objective and transparent in our methodology. The interviews have been coded independently by two researchers (RvE & LK). The resulting code book and themes have been discussed by the research group (RvE, LK, AB, WK, EK, MvE). Emerging themes have been described in the results and supported by extensive quotations from the interviews and discussed in the context of prognostic disclosure in life-limiting neurological disease. In the reporting of our study we adhered to the COREQ-checklist for qualitative research which contributes to the transparency and thoroughness of the methods used. We feel that the comments of reviewer 2 support the methodological transparency and complete reporting.

Reviewer 2 Report

Congratulations to the author team for a well-written manuscript reporting a comprehensive qualitative study. The addition of the COREQ strengthens the manuscript which was a pleasure to read. The detail in the manuscript is presented well.  The results diagram (Fig 1) was very helpful in understanding the results, as these are very detailed, with powerful quotes supporting the findings. Your findings take our understanding of delivery of diagnosis and prognosis information further forward - they are now two very different conversations with different timings. The findings show that timing of information is still an issue for ALS care, and that individual's reactions are not always predictable, especially when there is a gap between information delivered at diagnosis, and that discussed in personalised prognosis. It was interesting to see that physicians were also occasionally surprised by the outcomes of the prognostic tool.

I have only minor suggestions to improve the quality of the manuscript.

  1. Section 2.4.1 Interview guide. As the guide was based on your literature review, please give a little more detail here as to the sources. This need only be a sentence with citations of the key sources that supported the guide development as a summary/overview.
  2. Formatting of subheadings. In the reviewer copy of the manuscript, the subheadings do not stand out well from the text (this may be just a typesetting issue). As they guide the reader concisely through the findings, it is important they stand out from the paragraphs explaining them.

Author Response

We thank the reviewer for their comments, which we have addressed in the revised manuscript as discussed below.

Comments to the Author

Congratulations to the author team for a well-written manuscript reporting a comprehensive qualitative study. The addition of the COREQ strengthens the manuscript which was a pleasure to read. The detail in the manuscript is presented well.  The results diagram (Fig 1) was very helpful in understanding the results, as these are very detailed, with powerful quotes supporting the findings. Your findings take our understanding of delivery of diagnosis and prognosis information further forward - they are now two very different conversations with different timings. The findings show that timing of information is still an issue for ALS care, and that individual's reactions are not always predictable, especially when there is a gap between information delivered at diagnosis, and that discussed in personalised prognosis. It was interesting to see that physicians were also occasionally surprised by the outcomes of the prognostic tool. I have only minor suggestions to improve the quality of the manuscript.

  1. Section 2.4.1 Interview guide. As the guide was based on your literature review, please give a little more detail here as to the sources. This need only be a sentence with citations of the key sources that supported the guide development as a summary/overview.

Response: We thank the reviewer for pointing out this omission. The literature used for the interview guide was obtained from a systematic review conducted as part of an earlier study (van Eenennaam et al 2020). We have added the topics for our interview guide with the correct references based on the systematic review to section 2.4.1: The interview guide was formulated by two researchers (RvE, AB) and based on a literature review which was performed as part of an earlier study on the development of a communication guide [13]. Interview topics included information needs [18–26], difference in experiences between patients and caregivers [19, 21, 22, 24, 27], emotional impact and hope [18, 24, 26, 28, 29], and satisfaction with prognostic disclosure [18, 20, 23–25, 28, 30].” (See lines 112-115)

  1. Formatting of subheadings. In the reviewer copy of the manuscript, the subheadings do not stand out well from the text (this may be just a typesetting issue). As they guide the reader concisely through the findings, it is important they stand out from the paragraphs explaining them.

Response: We agree with the reviewer that the subheadings were not always clear. We think this will partly be resolved during the final typesetting, however, to make the different sections more clear we numbered the overarching themes in the text (See lines 174, 192, 234, and 244) and we made the subthemes bold (for example lines 175 and 179).